# Protease-Triggered Release of Stabilized CXCL12 from Coated Scaffolds in an Ex Vivo Wound Model

**DOI:** 10.3390/pharmaceutics13101597

**Published:** 2021-10-01

**Authors:** Sabrina Spiller, Tom Wippold, Kathrin Bellmann-Sickert, Sandra Franz, Anja Saalbach, Ulf Anderegg, Annette G. Beck-Sickinger

**Affiliations:** 1Institute of Biochemistry, Faculty of Life Sciences, Leipzig University, Brüderstr. 34, 04103 Leipzig, Germany; sabrina.spiller@uni-leipzig.de (S.S.); bellmann@uni-leipzig.de (K.B.-S.); 2Department of Dermatology, Venerology and Allergology, Leipzig University, Johannisallee 30, 04103 Leipzig, Germany; tom.wippold@medizin.uni-leipzig.de (T.W.); sandra.franz@medizin.uni-leipzig.de (S.F.); anja.saalbach@medizin.uni-leipzig.de (A.S.)

**Keywords:** biomaterial coating, protease-triggered protein release, chemotaxis, wound healing, ex vivo porcine organ culture model

## Abstract

Biomaterials are designed to improve impaired healing of injured tissue. To accomplish better cell integration, we suggest to coat biomaterial surfaces with bio-functional proteins. Here, a mussel-derived surface-binding peptide is used and coupled to CXCL12 (stromal cell-derived factor 1α), a chemokine that activates CXCR4 and consequently recruits tissue-specific stem and progenitor cells. CXCL12 variants with either non-releasable or protease-mediated-release properties were designed and compared. Whereas CXCL12 was stabilized at the N-terminus for protease resistance, a C-terminal linker was designed that allowed for specific cleavage-mediated release by matrix metalloproteinase 9 and 2, since both enzymes are frequently found in wound fluid. These surface adhesive CXCL12 derivatives were produced by expressed protein ligation. Functionality of the modified chemokines was assessed by inositol phosphate accumulation and cell migration assays. Increased migration of keratinocytes and primary mesenchymal stem cells was demonstrated. Immobilization and release were studied for bioresorbable PCL-*co*-LC scaffolds, and accelerated wound closure was demonstrated in an ex vivo wound healing assay on porcine skin grafts. After 24 h, a significantly improved CXCL12-specific growth stimulation of the epithelial tips was already observed. The presented data display a successful application of protein-coated biomaterials for skin regeneration.

## 1. Introduction

In the past decades, novel biomaterials that provide some “smart” functions have been discovered. This includes a combination of biodegradability with strength and elasticity without undesired host responses. Polyesters such as polycaprolactone (PCL) were initially identified to fill critical size bone defects [1,2] and to provide, besides mechanical integrity, also biodegradability and porosity, which facilitates cellular infiltration and vascularization [3]. In combination with novel fabrication techniques, many shapes can be produced that allow for tailor-made implant design [4]. Degradation time can be fine-tuned by copolymerization with glycolides or lactides [5] which allows for the transfer from bone to even cardiovascular implants and skin regenerative applications. Such polyesters are already used for skin therapies [6,7,8,9,10]. However, all biomaterials are prone to foreign body reactions resulting in inflammation and encapsulation of the device [11,12]. Smart biomaterials might induce guidance to injured tissue sites of desired cell types. Chemokines, such as CXCL12 which is also named stromal cell-derived factor 1α (SDF-1α), are among the most promising molecules in wound healing and tissue regeneration. CXCL12 signals by the G protein-coupled receptors CXCR4 and CXCR7 [13] and is a major player in homing and mobilization of stem cells and progenitor cells [14,15]. It promotes angiogenesis and neovascularization [16,17,18], both are important processes in wound healing. During skin regeneration, CXCL12 is a major chemokine attracting epidermal stem cells and promoting early steps of re-epithelialization prominently linked to proliferation and migration of keratinocytes in order to close the wound. CXCL12 treatment enhances epidermal stem cell migration and proliferation and accelerates wound healing in a rat model [19]. Thus, it is highly involved in various stages of wound healing including homeostasis, inflammation, and proliferation [20]. As CXCL12 is tightly regulated in a spatiotemporal manner and chemokine activity is linked to stable gradients, a specific release mechanism in combination with strong surface anchorage is suggested (Figure 1).

Here, we describe a complex strategy for the immobilization and controlled release of matrix metalloproteinase 9 and 2 (MMP9/MMP2)-resistant CXCL12 in combination with surface immobilization by covalent linkage to a l-3,4-dihydroxyphenylalanine (DOPA)-peptide derived from the mussel foot protein of the blue mussel (*Mytilus edulis*). Non-releasable and releasable compounds were tested for their biological activity in vitro, and the results are provided for a skin-mimetic ex vivo model. Here, CXCL12-coated polycaprolactone-*co*-lactide (PCL-*co*-LC) scaffolds led to an accelerated wound closure.

## 2. Materials and Methods

### 2.1. Peptide Synthesis

Peptides were synthesized by solid-phase peptide synthesis (SPPS) using the 9-fluorenylmethoxycarbonyl (Fmoc)/*tert*-butyl (tBu) protection group strategy in the 15 µmol scale as described previously [21]. Additionally, Fmoc-DOPA(acetonide)-OH was introduced manually in 2 equivalents (eq), biotin was coupled in 3 eq using 1-hydroxybenzotriazole (HOBt) and *N*,*N*′-diisopropylcarbodiimide (DIC), and Fmoc-NH-EG_2_-COOH (13 atoms) was coupled in 1.5 eq with 1-[bis(dimethylamino)methylene]-1*H*-1,2,3-triazolo[4,5-*b*]pyridinium 3-oxide hexafluorophosphate (HATU) and *N*,*N*-diisopropylethylamine (DIPEA). All other building blocks were used in 5 eq by manual coupling using HOBt and DIC. Automated peptide synthesis was used for elongation by the CXCL12^50−68^ and matrix metalloproteinase cleavage site (MMPCS) sequences. N-terminally coupled cysteine was used with N-terminal *tert*-butyloxycarbonyl (Boc) protection in order to allow for orthogonal cleavage of the 4,4-dimethyl-2,6-dioxocyclohex-1-ylidenethyl (Dde)-protecting group of the lysine side chain, which was removed with 3% hydrazine in *N*,*N*-dimethylformamide (DMF) by at least 10× for 10 min.

Final cleavage and deprotection of the peptides were accomplished after 3 h of incubation with trifluoroacetic acid (TFA) and triisopropylsilane (TIS)/H_2_O as scavenger (90:5:5 (*v*/*v*/*v*)). Precipitation and washing of the crude product was performed as described before [21]. Finally, peptides were restored in 20% acetonitrile (ACN), lyophilized, and subsequently purified on a Phenomenex Aeris Peptide XB-C18 column (100 Å, 5 µm, 250 × 21.2 mm) using linear gradients of 0.1% TFA/H_2_O and 0.08% TFA/ACN. Analytical identification of the peptides was achieved by matrix-assisted laser desorption/ionization time-of-flight mass spectrometry (MALDI-ToF MS; Ultraflex III MALDI-ToF/ToF; Bruker Daltonik, Bremen, Germany).

### 2.2. Protein Expression and Purification

The cDNA encoding the human CXCL12 wild type (wt) and CXCL12^1–49^ were cloned into the pTXB1 vector as described previously [22]. The introduced variations [V49A], [S4V], and a C-terminal GPLS elongation were introduced by site-directed mutagenesis with primers bearing the mutation intrinsically. Primers for the generation of mutations were purchased from *biomers.net* (Ulm, Germany) and were designed based on the *Escherichia coli* (*E. coli)* codon usage.

Proteins were expressed as fusion proteins with a mini-intein derived from the *Mycobacterium xenopi* gyrase A gene and a chitin-binding domain (CBD), an affinity tag that allows for chromatographic purification on chitin beads and subsequent cleavage to obtain the reactive protein thioester [23,24]. Expression and purification were performed analogous to previous protocols with some modifications as follows [22]. Cell lysis was carried out by FastPrep-24™ 5G (MP Biomedicals, Santa Ana, California, USA) using 5 g silica beads/3 L expression culture at 6 m/s for 40 s. After centrifugation of cell lysates (60 min, 12,000 × g, 10 °C), the inclusion bodies were washed twice with a column buffer (20 mM 4-(2-hydroxyethyl)-1-piperazineethanesulfonic acid (HEPES), 0.5 M NaCl, and 1 mM ethylenediaminetetraacetic acid (EDTA)) containing 0.2% Tween 20 (*v*/*v*) and twice with a column buffer containing 1 M NaCl instead of 0.5 M. The inclusion bodies were solubilized in a column buffer containing 8 M urea (5 mL/L expression culture) at 4 °C. The solubilized fusion protein was slowly diluted with a column buffer to 3 M urea and subsequently loaded on chitin beads (New England Biolabs, Frankfurt/Main, Germany). The target protein was cleaved off three times at 4 °C overnight with a column buffer containing 0.1 M dithiothreitol (DTT) or 0.2 M sodium 2-mercaptoethane sulfonate (MESNa), and 0.2% Tween 20 (*v*/*v*) as well as 3 M urea. In the case of DTT thioester, hydrolysis was performed at pH 10.0 and 4 °C under moderate stirring for 4 h. The protein was concentrated in an Amicon Ultra-15 filter device (MWCO 3000 Da; Merck, Darmstadt, Germany) at 4 °C and 5000 × g and subsequently purified by preparative reversed-phase high-performance liquid chromatography (RP-HPLC) on a Phenomenex Jupiter 5 u C18-column (300 Å, 5 µm, 250 × 21.2 mm) using linear gradients of 0.1% TFA/H_2_O and 0.08% TFA/ACN. An analysis was performed using MALDI-ToF MS, and pure fractions were pooled and lyophilized. All processes were monitored by sodium dodecyl sulfate polyacrylamide gel electrophoresis (SDS-PAGE) and analytical RP-HPLC. Notably, the initial methionine, which is not present in the mature human protein, is not cleaved by *E. coli*; thus, all generated variants bear an additional N-terminal methionine at position 0 [25].

### 2.3. Expressed Protein Ligation and Refolding

Protein fragments were restored in 0.1 M Na_2_HPO_4_ and 6 M guanidine hydrochloride, pH 6.0, and 100 mM MESNa as well as 5 mM tris(2-carboxyethyl)phosphine (TCEP) were added. Both fragments were combined and incubated for 30 min at pH 5.0. To start the native chemical ligation (NCL), the pH was shifted to 7.41 and the reaction was performed under stirring at room temperature under continuous monitoring by analytical RP-HPLC on a Phenomenex Aeris Widepore XB-C18 column (200 Å, 3.6 µm, 250 × 4.6 mm) using linear gradients of 0.1% TFA/H_2_O and 0.08% TFA/ACN. Analytical identification of the product was achieved by MALDI-ToF MS. The reaction was stopped by adding 8 mL of 10% acetic acid. Subsequent purification was carried out by preparative RP-HPLC on a Phenomenex Jupiter 5 u C18-column (300 Å, 5 µm, 250 × 21.2 mm) using linear gradients of 0.1% TFA/H_2_O and 0.08% TFA/ACN. Pure fractions were pooled and lyophilized.

Refolding of the chemokine was realized by rapid dilution as described previously [26]. Final purification of the refolded protein was performed by preparative RP-HPLC on a Phenomenex Jupiter 5 u C18-column (300 Å, 5 µm, 250 × 21.2 mm), applying linear gradients of 0.1% TFA/H_2_O and 0.08% TFA/ACN. Identity and purity were determined with electrospray ionization (ESI)-Orbitrap MS (Orbitrap Elite™; Thermo Fisher Scientific, Waltham, MA, USA) or MALDI-ToF MS, and analytical RP-HPLC, respectively. The protein concentration was determined by photometric measurement at 280 nm using the corresponding extinction coefficient of 8,730 M^−1^cm^−1^ for variants without the surface-binding peptide (SBP) sequence and 16,806 M^−1^cm^−1^ for variants containing the SBP anchor peptide.

### 2.4. PCL-co-LC Scaffold Preparation and Coating

Embroidered PCL-*co*-LC scaffolds were kindly provided by Prof. Dr. Stefan Rammelt (TU Dresden, Germany). These scaffolds have an open porosity of 87% with pores from 100 to 800 µm [1]. The surface was prepared by *n*-heptane treatment for 10 min three times, and the scaffolds were hydrophilized for 15 min in 50% methanol/1 M NaOH (1:1, (*v*/*v*)), and washed with distilled water until the pH was neutral. Coating of PCL-*co*-LC scaffolds was performed overnight under gentle shaking in a 96-well plate (TPP, Trasadingen, Switzerland) in a concentration-dependent manner or with 1 µM protein solution in 10 mM Tris buffer (pH 8.0) at room temperature. For semi-sterile cell culture handling, the scaffolds were sterilized beforehand two times for 15 min under UV light. The following day, the protein solution was removed and the PCL-*co*-LC scaffolds were washed either four times with Tris-buffered saline (TBS; 3 g/L tris(hydroxymethyl)aminomethane (Tris), 8 g/L NaCl, 0.2 g/L KCl, pH 7.6) before transferring them into new reaction vessels or three times with Dulbecco´s phosphate-buffered saline (DPBS) before placing them under sterile conditions into bottom wells of a 96-well Transwell migration plate (Corning^®^ HTS Transwell^®^ 96-well permeable supports, 5 µm pore size; Corning, Kaiserslautern, Germany).

### 2.5. MMP9 and MMP2 Activation and Digestion in Solution

Enzymes (100 ng/µL) were activated with 1 mM 4-aminophenylmercuric acetate (APMA; Merck, Darmstadt, Germany) at 37 °C and 200 rpm in the case of MMP9 for 24 h and in the case of MMP2 for 2 h according to the manufacturer´s protocol (R&D Systems, Minneapolis, MN, USA). Next, enzymes were diluted in the MMP-activation buffer (50 mM Tris, 150 mM NaCl, 10 mM CaCl_2_, and 0.05% Brij-35 (*w*/*v*)).

The protein was dissolved in the MMP-activation buffer, and 20 ng of the activated enzyme was added (10 µM protein and 0.2 ng/µL enzyme). Digestion was performed at 37 °C and 200 rpm. Samples were taken, and the reaction was stopped either with 5 mM EDTA for MALDI-ToF MS or by adding an equal volume of 2× Laemmli reducing buffer (62.5 mM Tris buffer pH 6.8, 25% glycerol (*v*/*v*), 2% SDS (*w*/*v*), 2% bromophenol blue (*w*/*v*), and 5% 2-mercaptoethanol (*v*/*v*)) prior to heating for 5 min at 95 °C for analysis by Tris/tricine SDS-PAGE according to the protocol by Schägger [27].

### 2.6. Binding and Release Assay

The PCL-*co*-LC scaffolds were prepared, coated, and washed as described above (see Section 2.4). Following this, binding studies were performed by a biotin-like ELISA as explained previously [28]. For the release experiments, the scaffolds were washed once with the MMP-activation buffer (50 mM Tris, 150 mM NaCl, 10 mM CaCl_2_, and 0.05% Brij-35 (*w*/*v*)), and subsequently 300 µL of the MMP-activation buffer as well as 60 ng of the activated MMP9 (0.2 ng/µL) were added to all samples except for the 0 h sample, which was not incubated with the protease. Each time point was separately analyzed in duplicates and incubated at 37 °C and 200 rpm. At the given time points, the supernatants were collected into new reaction vessels and, afterwards, 100 µL/well was loaded on 96-well hydrophilic immuno plates (MaxiSorp black 96-well plates; Thermo Fisher Scientific, Waltham, MA, USA) overnight at room temperature under slight shaking. A standard curve of recombinant [S4V]-[V49A]-CXCL12-GPLS in defined concentrations in the MMP-activation buffer was coated as well on the immuno plate in parallel. After coating, the immuno plates were washed four times with TBS supplemented with 0.1% Tween 20 (*v*/*v*) (TBS-T) and blocked for at least 30 min with 10% bovine serum albumin (BSA) in TBS. The bound protein was detected by rabbit polyclonal anti-CXCL12 IgG primary antibody (1:500; #ab9797, Abcam, Cambridge, UK) in 1% BSA/TBS-T for 1.5 h and mouse anti-rabbit IgG conjugated to horseradish peroxidase (HRP) secondary antibody (1:5000; #sc-2357, Santa Cruz Biotechnology, Dallas, TX, USA) in 1% BSA/TBS-T for 1 h. Detection after four washing steps with TBS-T was performed by the addition of 150 µL/well of the 3,3′,5,5′-tetramethylbenzidine (TMB) substrate for 3 min under shaking. The reaction was stopped by adding 25 µL of 1 M HCl, and 100 µL/well was transferred into a clear-bottom 96-well plate. Absorption at 450 nm was measured with a Tecan plate reader (Tecan Group, Männedorf, Switzerland). The data were normalized by setting values at 0 min to 0% and endpoint values of each data set to 100% protein release.

### 2.7. Isolation and Culture of Murine Mesenchymal Stem Cells (MSCs)

The Committee on Animal Welfare of Saxony (Germany) approved the animal protocols used in this study (T05/20, 5 November 2020). Murine mesenchymal stem cells (MSCs) were isolated as previously described [29]. In brief, femora and tibiae of C57Bl6/J mice were enzymatically digested using 26 U/mL of Liberase DL (Roche, Mannheim, Germany) for 2 h at 37 °C and 5% CO_2_. Cell suspension was filtered through a cell strainer (70 µm) and seeded in tissue culture flasks (#658175; Greiner Bio-one, Frickenhausen, Germany). The MSCs were cultured in minimum essential medium–alpha-modification (αMEM medium; Lonza, Basel, Switzerland) supplemented with 10% fetal bovine serum (FBS; PAN-Biotech, Aidenbach, Germany) and 1% penicillin/streptomycin (Anprotec, Bruckberg, Germany) at 37 °C and 5% CO_2_. The medium was replaced every three days.

### 2.8. Flow Cytometry Analysis of Murine MSCs

Cells were trypsinized (Tryp-LE, Thermo Fisher Scientific, Waltham, MA, USA) and counted, and a total of 5 × 10^5^ cells were used for the staining. First, the cells were fixated with 4% buffered paraformaldehyde for 5 min at RT, followed by two washes with DPBS, followed by permeabilization with 0.1% saponin (in DPBS/Gelafusal^®^ (Serumwerk Bernburg, Bernburg/Saale, Germany)) for 10 min at RT. The primary antibody (#bs-1011R; Bioss, Freiburg, Germany) was diluted 1:100 (10 ng/µL in 0.1% saponin/DPBS/Gelafusal^®^) and directly added to the cells for 45 min at 4 °C. After, two washes with DPBS/Gelafusal^®^ secondary antibody (1:200; goat anti-rabbit IgG, F(ab’)2-FITC, #sc-3839; Santa Cruz Biotechnology, Dallas, TX, USA) as well as direct label antibodies (CD45-PECy5.5; CD11b-PECy7; CD140a-APC; Sca1-PE) diluted in 0.1% saponin/DPBS/Gelafusal^®^ were added for 45 min at 4 °C. The cells were again washed two times with DPBS/Gelafusal^®^, and the cells were measured by flow cytometry analysis using a BD FACSLyric™ (BD Biosciences, Franklin Lakes, NJ, USA).

### 2.9. Immunofluorescence Staining of Murine MSCs

The cells were trypsinized and counted, and a total of 1x10^4^ cells were reseeded on glass slides (8 mm) and incubated overnight at 37 °C and 5% CO_2_ in αMEM medium supplemented with 10% FBS and 1% penicillin/streptomycin. The next day, the medium was removed and the cells were washed twice with DPBS and fixated with ice-cold methanol for 10 min at −20 °C, followed by two washes with DPBS. For permeabilization, the cells were incubated for 10 min at RT with DPBS/0.5% Tween 20. Afterwards, the cells were washed three times with DPBS/0.1% Tween 20 and then blocked with DPBS/0.1% Tween 20/3% BSA for 10 min at RT. The primary anti-CXCR4 antibody (10 ng/µL in DPBS/0.1% Tween 20/1% BSA; #bs-1011R, Bioss) was added overnight at 4 °C. The next day, the cells were washed three times and incubated with secondary antibody goat anti-rabbit IgG F(ab’)2-AlexaFluor546 (#A11030, Thermo Fisher Scientific, Waltham, MA, USA) and 4′,6-diamidino-2-phenylindole (DAPI; #D9542, Merck, Darmstadt, Germany) for 30 min protected from direct light at RT, followed by three washes with DPBS/0.1% Tween 20. For mounting, ProLong™ Gold (#P36934, Thermo Fisher Scientific, Waltham, MA, USA) was used, and pictures were taken using a Keyence BZ9000 (Keyence, Neu-Isenburg, Germany).

### 2.10. Transwell Migration Assays

#### 2.10.1. Jurkat Cell Migration towards Coated or Uncoated PCL-*co*-LC

Migration assays with coated or uncoated PCL-*co*-LC scaffolds were performed as described previously [30]. In brief, the materials were placed in the bottom chamber of a 96-well Transwell migration plate (Corning^®^ HTS Transwell^®^ 96-well permeable supports, 5 µm pore size; Corning) in 250 µL of the migration medium (RPMI 1640 + 2% FBS + 2 mM CaCl_2_) and 0.2 ng/µL activated MMP9 in the MMP-activation buffer or buffer without protease were added. In the upper chamber, 1 × 10^5^ Jurkat cells per well were seeded in 100 µL migration medium. Following incubation for 4 h at 37 °C, 100 µL of the cell suspension from the bottom chamber were mixed with 100 µL of trypan blue (0.5% (*w*/*v*) in DPBS), incubated for 1 min and counted using Tecan cell counting chips with a Tecan Spark^®^ multimode microplate reader (Tecan Group, Männedorf, Switzerland). Blank materials were used for data normalization. Migrated cells towards bare materials were set to 1, and the data are presented as x-fold over control as the mean ± standard deviation (SD) from at least three independent experiments.

#### 2.10.2. Concentration-Dependent Jurkat Cell Migration

For concentration-dependent migration assays, the dilution series of tested CXCL12 variants were prepared in migration medium and 250 µL were transferred into the bottom chamber. Subsequently, 1 × 10^5^ Jurkat cells per well were seeded in the upper chamber in 100 µL migration medium and the cells were incubated at 37 °C for 2 h. Seeded cells (1 × 10^6^ cells/mL) were set to 100% migrated cells. Cell measurement was performed as explained above (see Section 2.10.1.). The data are presented as the mean of the percent migrated cells ± SD from at least three independent experiments.

#### 2.10.3. Concentration-Dependent Migration of Murine MSCs

Isolated and characterized murine MSCs were trypsinized, counted, and a total of 1.8 × 10^5^ cells were seeded into a ThinCert™ insert (#662638; Greiner Bio-one, Frickenhausen, Germany) and placed into a 24-well plate. A total of 500 µL of the respective stimulants in αMEM medium with 1% FBS (50 ng/mL platelet-derived growth factor (PDGF-AB, #315-17; Peprotech, NJ, USA); 100 nM CXCL12 wt; and 100 nM or 500 nM compound **4**) were placed in the vessel bottom, and the cells were incubated for 5 h at 37 °C and 5% CO_2_. After incubation, the inserts were carefully removed from the plate. The cells that migrated through the insert into the vessel were harvested and counted (medium flow speed for 90 s) by flow cytometry analysis using a BD FACSLyric™ (BD Biosciences, Franklin Lakes, NJ, USA). The results were normalized to 1% FBS-mediated migration and analyzed using GraphPad Prism5 v5.03 software (Graph-Pad Software, San Diego, CA, USA). The data were obtained in triplicates from two independent experiments and are shown as mean ± SD.

### 2.11. Inositol Phosphate Accumulation Assay

CXCR4 cDNA C-terminally fused to eYFP sequence was cloned into pVitro2 vector by standard polymerase chain reaction using the Mlu I and Sal I restriction sites. Correct sequences and open reading frames were confirmed by Sanger DNA sequencing. For measuring receptor activation using the Gα_q_ pathway, the untagged chimeric Gα_Δ6qi4myr_ protein was used in pcDNA3.1 vector, which was previously described [31] (kindly provided by Prof. Dr. Evi Kostenis, Rheinische Friedrich-Wilhelms-Universität, Bonn, Germany). Next, COS-7 cells (African green monkey) were cultured in Dulbecco’s modified Eagle’s medium (DMEM) with 4.5 g/L glucose and l-glutamine (Lonza, Basel, Switzerland) supplemented with 10% heat-inactivated FBS (Lonza, Basel, Switzerland) at 37 °C in humidified atmosphere and 5% CO_2_ and were transiently transfected for 6 h in 25 cm^2^ cell culture flaks with 3 μg of pVitro2-CXCR4_eYFP plasmid and 1 μg of the plasmid containing the chimeric G protein Gα_Δ6qi4myr_ using Metafectene^®^ Pro (Biontex Laboratories, Munich, Germany). After transfection, the cells were reseeded into a white 384-well plate (1 × 10^4^ cells/well; Greiner Bio-one, Frickenhausen, Germany) and cultured for at least 16 h. Inositol phosphate (IP) accumulation assay was performed as previously described [32] using the IPone Gq assay kit (PerkinElmer/Cisbio Bioassays, Codolet, France). Stimulation has been performed for 60 min. The results were normalized to minimum/maximum recombinant CXCL12-mediated CXCR4 activation and analyzed using GraphPad Prism5 v5.03 software (Graph-Pad Software, San Diego, CA, USA). The data were obtained from at least three independent experiments and is shown as mean ± standard error of the mean (SEM).

### 2.12. Analysis of HaCaT Cell Migration

HaCaT cells (7 × 10^4^ cells per well) were seeded in an Incucyte^®^ ImageLock 96-well plate (Essen BioScience, Ann Arbor, MI, USA) and grown in DMEM (Anprotec, Bruckberg, Germany) supplemented with 10% FBS (PAN-Biotech, Aidenbach, Germany) and 1% Zellshield (Biochrom, Berlin, Germany). The following day, 10 µg/mL mitomycin C (Merck, Darmstadt, Germany) was added, and after 3 h, the medium was changed to DMEM supplemented with 0.5% FBS. A 700–800 µm wide scratch was applied in the cell-monolayer with IncuCyte^®^ WoundMaker (Essen BioScience, Ann Arbor, MI, USA), and the remaining cells were washed twice with DPBS (Anprotec, Bruckberg, Germany). Recombinant CXCL12 wt and its variants in DMEM with 0.5% FBS (100 nM final protein concentration) were added to the cells. The cells were incubated for up to five days, and migration was determined as closed part of initial wound area every 2 h with IncuCyte^®^ scratch wound cell migration software module (Essen BioScience, Ann Arbor, MI, USA).

### 2.13. Analysis of CXCL12 Signaling

Human fibroblast cultures were obtained from human female breast skin (approval was obtained from the local Ethics Committee No. 363-12-051120 on 5 November 2020). The fibroblasts were used at passages 2–3, when 1.5 × 10^5^ cells per well were seeded in 6-well plates and incubated at 37 °C and 5% CO_2_ with DMEM/10% FBS until the cells were almost confluent, followed by incubation overnight with DMEM/0.5% FBS. The cells were washed once with DPBS and incubated for 10 min with recombinant CXCL12 wt or its variants in DMEM (10 ng/mL final protein concentration). The supernatants were completely removed and the cells were washed once with cold DPBS on ice. The cell extracts were prepared by adding 50 µL of the SDS-reducing buffer (Cell Signaling Technologies, Danvers, MA, USA) containing 2-mercaptoethanol (0.1% (*v*/*v*)). Fifteen microliters of lysate per lane was separated by SDS-PAGE (TGX Stain-Free™ Protein Gels 4–20%; Bio-Rad, Hercules, CA, USA) and blotted on polyvinylidene difluoride (PVDF) membranes (Immobilon^®^-P, 0.45 µm; Merck, Darmstadt, Germany). The activated pAkt was detected by a rabbit anti-phosphoAkt (Ser473) primary antibody (#4060; Cell Signaling Technologies, Danvers, MA, USA). Total Akt was detected after stripping with a rabbit anti-Akt antibody (#4691; Cell Signaling Technologies, Danvers, MA, USA). Activated pErk1/2 was detected by a rabbit anti-phosphoErk1/2 (Thr202/Tyr204) primary antibody (#9101; Cell Signaling Technologies, Danvers, MA, USA). Goat anti-rabbit IRDye^®^ 680RD and goat anti-rabbit IRDye^®^ 800CW (LI-COR Biosciences, Lincoln, NE, USA) were used as secondary antibodies. The blots were visualized using Odyssey Fc Imaging System (LI-COR Biosciences, Lincoln, NE, USA). The expression of glyceraldehyde 3-phosphate dehydrogenase (GAPDH; #AB2302, Merck, Darmstadt, Germany) and total Akt were used as controls.

### 2.14. Organ Culture of Porcine Skin

Ears from freshly sacrificed pigs were obtained from the Medical-Experimental Center of Leipzig University. The organ culture was set up according to a method described by Brander et al. [33] and modified slightly. A detailed protocol was described previously [34]. Additionally, PCL-*co*-LC scaffolds were coated with releasable and non-releasable CXCL12 variants as described above under semi-sterile conditions (see Section 2.4). After a washing step, the scaffolds were placed directly on the wounds overlapping the wound margins. Uncoated scaffolds were used as controls. Solute CXCL12 wt was applied using filter papers. Therefore, filter papers with a diameter of 6 mm were placed into 30 µL of a 10 µM CXCL12 wt solution until the complete volume was soaked up, and the paper were placed on the wounds. For blocking CXCR4 activity, 50 ng/mL AMD3100 (#239820; Merck, Darmstadt, Germany) was added. The cultures were incubated at 37 °C for up to 48 h. The wounded skin was fixed in 4% buffered paraformaldehyde and embedded in paraffin.

### 2.15. Histological Analysis

For histological examination, dewaxed and rehydrated sections (6 µm thickness) were used. Sections were stained with MassonGoldner-Trichrome according to the manufacturer’s protocol (Carl Roth, Karlsruhe, Germany) and mounted with Entellan^®^ (Merck, Darmstadt, Germany). Pictures were taken and stitched using a Keyence BZ9000 microscope (Keyence, Neu-Isenburg, Germany). The length of newly formed epidermis tips was measured with ImageJ v1.51k software (National Institutes of Health, Bethesda, MA, USA) on both wound sites, and 8–12 wounds were analyzed per condition.

### 2.16. Statistical Methods

The data were analyzed from at least two or three independent experiments as stated using GraphPad Prism5 v5.03 software or GraphPad Prism8 v8.4.3 software, respectively. Visualization was performed either as mean ± SD or as mean ± SEM as indicated. Statistical analysis was conducted for the release assay by an unpaired *t*-test comparing differences between groups, and a Jurkat cell migration assay towards the coated material was evaluated using one-way analysis of variance (ANOVA) with Tukey’s post hoc test. Keratinocyte migration assay was analyzed by two-way ANOVA, and Transwell migration assay with murine MSCs as well as ex vivo porcine epidermal tip length were analyzed by one-way ANOVA with Fisher’s Least Significant Difference (LSD) post hoc test. *p* values ≤ 0.05 were considered statistically significant.

## 3. Results

### 3.1. Generation and Signal Transduction of CXCL12 Variants with Stable or Protease-Mediated Release Linker

In order to investigate the functionality of releasable CXCL12, we designed and synthesized CXCL12 wt and six variants, N-terminally modified by an [S4V] exchange to provide protease stability, and C-terminally extended by various linkers to immobilize the protein to surfaces and to allow for protease-mediated release (Table 1). The sequence details and properties of the generated CXCL12 variants are described in Appendix A. Full-length CXCL12 and its analogs (Table 1; wt and compounds **5**–**6**) were expressed in *E. coli* ER2566, whereas the modified variants (Table 1; compounds **1**–**4**) were obtained by expressed protein ligation (EPL). For these variants, the N-terminal fragment was expressed in *E. coli* ER2566 and purified by the IMPACT™-system, resulting in a reactive protein thioester. The C-terminal fragment with a native cysteine at the N-terminus was synthesized by SPPS in combination with the SBP and a three-fold repetitive protease cleavage site, which can be cleaved by MMP9 and MMP2. This cleavable linker was previously described and characterized [35]. Both fragments were ligated by native chemical ligation (NCL), resulting in a native peptide bond (Appendix A). After refolding of the chemokine by rapid dilution (Appendix A) and subsequent purification by preparative RP-HPLC, the identity of all variants was proven by MALDI-ToF MS, and purity (>95%) was demonstrated by analytical RP-HPLC (Table 1D,E and Appendix A). Furthermore, the functionality of each variant was tested in an IP accumulation assay (Table 1F and Appendix A) and displayed a slight loss in activity at the CXCR4, but all analogs are still active in the nanomolar range. Valine at the very C-terminus of the N-terminal fragment at position 49 was exchanged by alanine ([V49A]-CXCL12^1−49^), which is known to be more reactive in transthioesterification reactions [36,37,38]. Since the N-terminus of the chemokine is prone to protease degradation and subsequent biological inactivation, serine at position 4 was replaced by valine ([S4V]-[V49A]-CXCL12) which resulted in stabilized CXCL12 variants. All generated variants as well as CXCL12 wt are stable against the ubiquitously expressed dipeptidyl peptidase 4 (DPP4) due to the additional methionine at position 0, which is not removed by *E. coli* after expression in the case of sterically demanding N-terminal amino acids such as lysine.

Several cell types in the skin express CXCL12 and/or its cognate receptor CXCR4, including MSCs, fibroblasts, pericytes, endothelial cells, and infiltrating immune cells [39,40]. Therefore, we tested the efficiency of our modified CXCL12 compounds exemplarily on MSCs, fibroblasts, a keratinocytes cell line (HaCaT), and a T-lymphocyte cell line (Jurkat).

We used the unmodified human CXCL12 wt as a positive control and the MMP-resistant, releasable, and surface-adhesive CXCL12 variant compound **4** as the derivative designed for the biomaterial interface. Compound **5** was used to mimic the released MMP-stable CXCL12 generated from compound **4** after linker cleavage by MMP9 or MMP2. In order to test the influence of the C-terminal residual amino acids GPLS after release from the surface, we also investigated compound **6** without any C-terminal elongation as a control.

Jurkat cells, which express the CXCR4 constitutively, were used to identify biologically active concentrations and showed a migration maximum at 700 nM for the releasable compound **4** in a Transwell migration assay (Figure 2A). CXCL12 wt showed the best response at 30 nM. Compound **4** still induced cell migration, albeit at higher concentrations, whereas compound **5** was able to regain migration potential, mimicking the released CXCL12 form after MMP9 or MMP2 cleavage. This chemokine version should be active in wounds after release by forming chemokine gradients. No further improvement was detected without any C-terminal linker of compound **6**. Primary murine MSCs (CD45^neg^, CD11b^neg^, Sca1^pos^, and CD140a^pos^) were tested for CXCR4 expression to ensure sensitivity to CXCL12 variants (Appendix A). Murine MSCs responded similar to CXCL12 wt and compound **4** with a concentration of 100 nM in the bottom well in a Transwell migration assay but showed a significantly elevated migration to a higher concentration of compound **4** at 500 nM (Figure 2B). A human keratinocyte cell line (HaCaT cells) responded similar to 100 nM CXCL12 wt and compound **4** in a gap closure scratch assay over five days (Figure 2C and Appendix A). The cells were pre-treated with mitomycin C, a potent proliferation inhibitor. In consequence, the observed effect is specific to the migration of the cells and does not result from cell division. Specific activation of the PI3K-Akt signaling pathway and the Ras-Raf-Erk1/2 pathway as direct migration pathways were investigated in human skin fibroblasts (Figure 2D). The cells were treated with 100 nM CXCL12 wt or compound **4** for 10 min, and elevated levels of phosphorylated Akt and Erk1/2 in a Western blot analysis have been detected as an indicator for pathway activation. In summary, several human and murine cell types, which are important during wound healing, responded to the CXCL12 analogs.

### 3.2. Protease-Induced Surface Release of Compound ***4*** and Cellular Response to Gradient Formation

As we observed a good activity profile of our generated constructs, we tested the coating of PCL-*co*-LC scaffolds in a concentration-dependent manner with the biotinylated CXCL12 variants compounds **1** and **2** and used the biotinylated anchor peptide SBP as control (Appendix A). Full coating of the PCL-*co*-LC scaffolds was achieved for the longest version compound **2** of a concentration of 480 nM.

Next, it was our aim to validate our concept of protease-mediated protein release from surfaces. MMP2 and MMP9 are important proteases in the wound tissue [41,42], and wound fluids can be generated thereof. Therefore, we proved the general functionality of the protease-cleavable linker of compound **4** in solution. Rapid truncation of the chemokine N-terminus was demonstrated for CXCL12 wt by Tris/tricine SDS-PAGE (Appendix A) and by MALDI-ToF MS (Figure 3A, M_calc_ intact = 8090 Da, M_calc_ truncated = 7547 Da), whereas for compound **4** the N-terminal stabilization was verified and the efficient cleavage of the linker sequence by MMP9 as well as by MMP2 was investigated (Figure 3B and Appendix A). After 15 min partial cleavage of the three-fold repetitive linker sequence has been observed (Appendix A, 9.7 kDa, 9.1 kDa, and 8.4 kDa) and at least after 60 min, the linker was fully chopped right after the first cleavage site at the C-terminus of CXCL12 (M_calc_ intact = 11,572 Da, M_calc_ cleaved = 8429 Da). Finally, PCL-*co*-LC scaffolds were coated with compound **4** and incubated with or without MMP9 at 37 °C. Release was investigated by ELISA against CXCL12 (Figure 3C). Similarly, treated uncoated surfaces were used as controls and obtained values subtracted as a background. A burst release was detected within the first hours with fast kinetics (t_1/2_ = 0.85 h, k = 0.81 ± 0.23 h^−1^). No unspecific release was observed without MMP9, which highlights the strong surface binding and the stimuli-responsive release strategy. In order to investigate cellular response to our coatings, we performed Transwell migration assays with Jurkat cells. Uncoated and coated materials with either compound **3** or compound **4** were placed in the bottom chamber of a Transwell migration plate with or without the addition of MMP9 and migrated Jurkat cells were counted (Figure 3D). The data were normalized to uncoated materials and coating with compound **3**, lacking the MMP cleavage site, was used as a control. A significantly increased cell number responded to the released compound **4** with MMP9 treatment of the coated PCL-*co*-LC scaffolds compared with uncoated scaffolds.

### 3.3. Ex Vivo Wound Closure Model Using Porcine Organ Culture

Human and porcine CXCL12 (97%) and CXCR4 (94%) share a high-sequence homology (Appendix A). Therefore, PCL-*co*-LC scaffolds were coated with human CXCL12 variants and placed on skin wounds in an ex vivo porcine organ culture setup for 48 h. Epithelial tip lengths were measured as an indicator for initiation of wound closure (Figure 4). We applied the soluble CXCL12 wt by filter papers instead of topical application in order to prevent the protein solution from running off the wound area. Coating with compound **4** resulted in a significantly improved wound closure for each time point. Furthermore, this effect was CXCL12-specific because the addition of AMD3100, a potent CXCR4 inhibitor, blocked the effect. Interestingly, PCL-*co*-LC scaffolds coated with MMP-releasable compound **4** were as efficient as filter papers soaked with soluble CXCL12 wt at early time points of treatment. Moreover, the non-releasable compound **3** resulted in lower responses, suggesting that the release of intact CXCL12 from compound **4** by MMPs in these wounds was essential. Since the setup with AMD3100 did not show any increased tip lengths in our model, CXCR4 signaling is involved in tip elongation supporting our hypothesis.

## 4. Discussion

Wound healing is a complex process divided into four major overlapping phases: hemostasis, inflammation, proliferation, and remodeling [43,44]. First, thrombocytes reach the wounded area and a fibrin clot is formed to prevent blood loss. Second, neutrophils and macrophages enter the wound site to remove tissue debris and to prevent infection, followed by the recruitment of stem cells, fibroblasts, endothelial cells, and keratinocytes in order to close the wound. Angiogenesis is an important process during both, inflammation and proliferation, assisting cellular invasion towards the wound. In the final stage, myofibroblasts reorganize the collagen matrix. All stages must be orchestrated by modulator molecules such as chemokines that play key roles in the regulation of angiogenesis and recruitment of inflammatory as well as stem cells and progenitor cells [45]. A disruption of these finely organized processes can lead to chronic and/or non-healing wounds. Various common diseases such as diabetes mellitus, obesity, and hypertension promote the development of chronic and non-healing wounds [46]. For treatment of those, some Food and Drug Administration (FDA)-approved wound dressings targeting chronic wounds, are on the market. This includes antimicrobial silver-containing skin dressings (e.g., Aquacel^®^ Ag, ConvaTec, Deeside, UK), cross-linked bovine collagen with glycosaminoglycans (GAGs) (Integra^®^ bilayer matrix, Integra Lifesciences, Plainsboro, NJ, USA), skin grafts containing living stem cells, keratinocytes, and fibroblasts, mimicking functions of healthy skin (Apligraf^®^, Organogenesis, Canton, MA, USA) as well as gels equipped with recombinant human PDGF (Regranex^®^ gel, Smith & Nephew, London, UK) [47,48]. The latter is the only approved growth factor used in the treatment of diabetic foot ulcers, suggesting the effectiveness of delivering a single modulator [49,50]. However, application of Regranex^®^ has been linked with elevated cancer risk [51].

### 4.1. CXC12 for Treatment of Disturbed Wound Healing

The involvement of CXCL12 and CXCR4 in the regeneration of various tissues and some applications of CXCL12 in regenerative medicine addressing stem cells and progenitor cells have been summarized previously [52]. CXCL12 is an important wound healing promoter and a potent chemoattractant for stem cells and progenitor cells [53,54] as well as an initiator of keratinocyte migration and proliferation [55,56,57]. In acute wounds, CXCL12 expression levels are markedly upregulated [39,58], whereas in chronic wounds, similar to diabetic foot ulcers, CXCL12 was found to be downregulated resulting in disturbed endothelial progenitor cell homing and delayed wound healing [59,60]. While the complex pathophysiology of chronic wounds is still unclear, the treatment with CXCL12 was shown to rescue wound healing in different animal models [61,62,63,64].

Major amounts of CXCL12 are produced post-wounding by fibroblasts rather than by keratinocytes, which instead express the CXCR4 [39,55,57]. Accordingly, we observed a migratory response of HaCaT cells in a scratch assay, where CXCL12 wt as well as our tested compound **4** led to similar results in solution (Figure 2C). Fibroblastic CXCL12 expression provides guidance for wound healing-related cells and proliferation of keratinocytes in cutaneous wound healing [55,56]. Fibroblasts may also respond to CXCL12 in an autocrine manner [65], and we have shown CXCR4 signaling response to the wt chemokine and the modified compound **4**, respectively (Figure 2D). Thus, we have demonstrated that CXCL12 and the synthesized variants thereof show the desired biological activity in cells that are relevant for wound repair.

### 4.2. Immobilization of CXCL12 on Biomaterials with Gradient Formation

Enhancing the CXCL12/CXCR4 signaling axis in biomedical applications particularly needs a proper delivery platform. One opportunity is the equipment of tissue-contacting devices with such wound healing mediators. The coating of biomaterial surfaces should lead to a beneficial wound healing process and furthermore reduce the foreign body reaction. A chemotactic gradient that guides tissue specific stem cells and progenitor cells to injured sites is suggested. Wound healing-mediating proteins are often applied by affinity-based electrostatic interactions, predominantly mediated by heparin or other sulfated GAGs [66,67,68], whereas some approaches utilize cross-linking chemistry for covalent linkage [69,70,71,72]. However, controlled and specific covalent protein immobilization is less frequently used [35,73]. Diverse immobilization strategies for growth factors and cytokines have been greatly reviewed previously [74,75]. Heparin-based PEG hydrogels loaded with CXCL12 are often applied [76,77,78], and hydrogels packed with CXCL12 mutants have been proven to attract early endothelial progenitor cells with a long-term release profile upon mutations of the heparin binding site of CXCL12 [26,79]. Other non-covalent approaches used a heparin-based sandwich-like setup, where the chemokine is bound by soluble heparin, which in turn is immobilized by a heparin-binding peptide [30,80]. Here, release kinetics might be more complex due to diffusion of CXCL12 alone or in combination with heparin. However, release curves displayed a similar shape with a slower kinetic compared to the enzyme-triggered release of our approach (Figure 3C) [30]. Nonetheless, the enzymatic release might be prolonged in vivo due to lower physiological MMP levels. Accordingly, we observed an increased migration response of Jurkat cells towards a CXCL12 gradient produced in situ by MMP9 from coated PCL-*co*-LC scaffolds (Figure 3D) as this was shown from polymer films and was also seen for heparin-mediated release of CXCL12 in previous studies [30,35].

Focusing on the bio-functional coating of materials, our approach enables a triggered release of the active protein from the biomaterial surface as a response of MMP9 and MMP2 action (Figure 3B,C). Therefore, we approved the efficient binding of our anchor peptide to PCL-*co*-LC scaffolds (Appendix A) as it was shown for other DOPA-containing binding peptides [30,81,82]. The surface adhesion is realized solely by the incorporation of DOPA, a versatile surface adhesive post-translationally modified amino acid found in the mussel foot protein of the blue mussel (*Mytilus edulis*). Synthetically derived DOPA-containing peptides or peptidomimetics have been engineered as coatings for modification of biomaterials, effectively utilizing the stable binding of mussel proteins [83,84]. On a different note, DOPA-functionalized hydrogels can be used as tissue glue by cross-linking cellular proteins [85]. The versatile DOPA chemistry was summarized before [86]. Previous studies certified the DOPA-mediated binding to several other surfaces such as aluminum oxide, polystyrene, glass, stainless steel, PTFE, as well as nitinol [30,87,88], thereby broadening the application range from bone implants to cardiovascular devices.

### 4.3. MMPs of the Wound Milieu May Be Used for Controlled Release of CXCL12 from Engineered Biomaterials

Further factors in wound healing are MMPs and their tight regulation matters. However, MMPs and other proteases are overexpressed in chronic wounds [42,59,89], being capable of inactivating modulator proteins such as CXCL12 [59,90]. Active MMP2 and MMP9 were increased 6-fold and 14-fold, respectively, in lesions of diabetic patients compared to non-diabetic conditions [42]. These highly abundant proteases in wounded tissue act as regulators during the wound healing cascade by activating different proteins in inflammatory stages and are involved in remodeling of the extracellular matrix in later wound healing stages [91,92,93,94], thereby enabling cell migration [95]. MMP9 and MMP2 are two well-characterized members of a family of 25 zinc-dependent endopeptidases, which are able to degrade or cleave a wide range of substrates [96,97,98].

Here, we aimed to use the endogenous expression of MMP9 and MMP2 for triggering an effective CXCL12 release on demand. We incorporated a MMP9/MMP2-sensitive peptide linker at the C-terminus of CXCL12 enabling the release in the desired time frame. This cleavage site was previously characterized for MMP9 [35]. However, both MMP9 and MMP2 are able to inactivate CXCL12 by N-terminal truncation [90] resulting in a neurotoxic remnant CXCL12^5−68^ [99]. Due to the introduction of a protease-cleavable linker, it is essential to use protease-resistant chemokines to avoid the release of truncated and thereby inactivated proteins (Figure 3A) [59,90]. Of note, already one amino acid exchange at the N-terminus led to CXCL12 stable against MMP9 and MMP2 as demonstrated by Segers et al. [100]. In 2007, the CXCL12 mutant [S4V]-CXCL12 was identified which is MMP9/MMP2-resistant but retains chemotactic activity [59,100,101]. We combined this N-terminal stabilization of CXCL12 with the directed enzymatically cleavable linker in order to release the active chemokine in the correct time frame (Figure 3B) [35,101]. Other MMP-sensitive linkers were incorporated in PEG hydrogels modulating degradability for improved chondrogenesis [102] or were used for masking of cytokines which were released and activated at the site of disease to overcome pharmacokinetic issues [103]. As MMPs are also overexpressed in cancer [104], they became versatile agents in tumor targeting by MMP-responsive drug delivery [105].

CXCL12 was often incorporated into GAG-containing hydrogels [76,78,79] and MMP activity was used for hydrogel degradability [102,106,107,108]. Most of these studies dealt with non-stabilized CXCL12, few used the [S4V] stabilization in combination with biomaterial applications [35,77]. However, using a library approach the stabilized CXCL12 was found to be 3-fold less potent for migration of Jurkat cells, and a 16-fold loss in potency in cAMP signal transduction assays compared with CXCL12 wt was detected [100]. Others found a four-fold decrease in potency in IP accumulation assays [35]. In our studies, we found similar results and detected an 11-fold decrease in migration (Figure 2A) and a consistent 10-fold loss in potency in IP accumulation assays of compound **6** compared with CXCL12 wt (Table 1 and Appendix A). Notably, we additionally exchanged valine at position 49 in order to improve reaction kinetics during chemokine synthesis [36,37,38], although this residue was predicted to interact with the CXCR4 in a hydrophobic pocket of CXCR4 residues, Thr13 and Met24 were consequently shown to be involved in chemotactic activity of CXCL12 [109,110]. Nevertheless, Baumann et al. [22] used the amino acid exchange to alanine for fast fragment coupling in EPL and were able to show neither that Val49 is necessary for proper function of CXCL12 nor that the mutation disturbs the overall structure. They found similar results for the mutant as for the CXCL12 wt in IP accumulation assays as well as in Jurkat migration assays. This is in agreement with our data, as we detected only a slight loss of activity by introducing the [V49A] mutation (compound **1**, Table 1F) independent of the C-terminal elongation by the SBP sequence. By addition of the [S4V] mutation, we found a further loss in activity in signal transduction assays (compound **3**, Table 1F) which is kept stable independent of shorter or longer C-terminal modifications. However, this is not the case for migration assays, where a longer C-terminal elongation (compound **4**, Figure 2A) led to a dramatic loss in migration potential compared to the variant that mimicked the released CXCL12 (compound **5**, Figure 2A). Accordingly, primary murine MSCs also showed an elevated response to a higher concentration of compound **4** (Figure 2B). Consequently, migration potential was regained by two-fold after linker cleavage (Figure 2A). The different responses in IP accumulation assays and migration assays are likely to be due to occupation of different signaling pathways as the migration response is a very complex process [111]. Furthermore, it is speculated that this might be an indicator for biased signaling [112,113].

### 4.4. Functional Testing of CXCL12-Modified Biomaterials in an Ex Vivo Wound Healing Model

Finally, ex vivo wound healing studies were performed to gain an impression on wound repair potential of our releasable CXCL12 variant. CXCR4 expression was detected in proliferating epithelial cells [39], and several studies on cutaneous wound healing demonstrated positive effects of CXCL12 on wound closure. The skin culture model allows the analysis of short term effects of biomaterials on wound-resident cells but excludes the effects from immigrating cells from the circulation. Thus, we focused on the analysis of migration and putative proliferation of epithelial keratinocytes measured as an epidermal tip length. This parameter was significantly influenced by CXCL12 released from the PCL-*co*-LC scaffolds (Figure 4).

Other studies demonstrated accelerated wound closure in rats after intradermal CXCL12 injection in combination with increased endothelial cell numbers [19]. On the other hand, CXCL12/CXCR4 signaling facilitated wound healing in mice by recruiting bone marrow-derived MSCs to the wound site in addition to stimulating local cell migration [114] which is in accordance to our in vitro Transwell migration assay with murine primary MSCs (Figure 2B). In both studies, the effects were blocked by the addition of AMD3100, a potent and specific CXCR4 inhibitor [55,115].

Our approach switched previous affinity-based and cross-linking immobilization systems to an enzyme-triggered release setup with a covalent and oriented immobilization strategy. We observed a significantly improved wound closure for the releasable compound **4** compared with uncoated materials and scaffolds coated with the non-releasable compound **3** (Figure 4B). Importantly, we demonstrated that this effect was CXCL12-specific as we blocked increased wound closure by the addition of AMD3100. Additionally, the applied CXCL12 wt by soaked filter papers showed similar short term effects in this organ culture model. However, an improved and more persisting action of compound **4** might be expected in vivo, when the chemokine is actively released for longer time periods. This scenario cannot be investigated with the tissue culture model due to tissue degradation after 50–60 h of incubation and will be addressed in future studies. Moreover, the ex vivo porcine model did not involve the action of inflammatory cells (neutrophils and infiltrating macrophages), a major source of enzymes such as elastase and MMPs. In non-healing wounds, we found a prolonged infiltration of inflammatory cells and their mediators including tissue degrading enzymes. Thus, we expect that compound **4** might prolong the release of active CXCL12 in a classical wound environment.

In summary, bio-functional coatings of scaffolds with immobilized and releasable CXCL12 upon a given enzymatic stimulus activate MSCs, keratinocytes, and fibroblasts as well as wound repair. These insights may be applied in future for skin regenerative approaches.

## Figures and Tables

**Figure 1 pharmaceutics-13-01597-f001:**
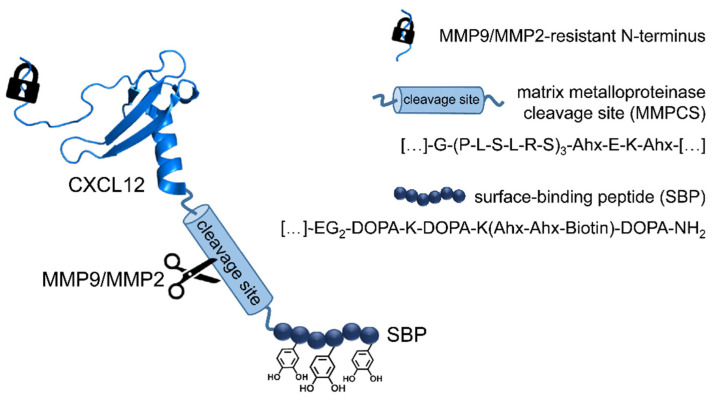
Conceptional scheme. Stabilized CXCL12 (PDB: 2KEE) is equipped with surface adhesive properties such as l-3,4-dihydroxyphenylalanine (DOPA) and a protease cleavable linker (MMPCS). Upon a given stimulus, active CXCL12 can be released from the surface by enzymatic cleavage by matrix metalloproteinase 9 and 2 (MMP9 and MMP2). The generated chemokine gradient results in attraction of the desired cell types towards the biomaterial interface. Ahx = 6-aminohexanoic acid; EG = ethylene glycol.

**Figure 2 pharmaceutics-13-01597-f002:**
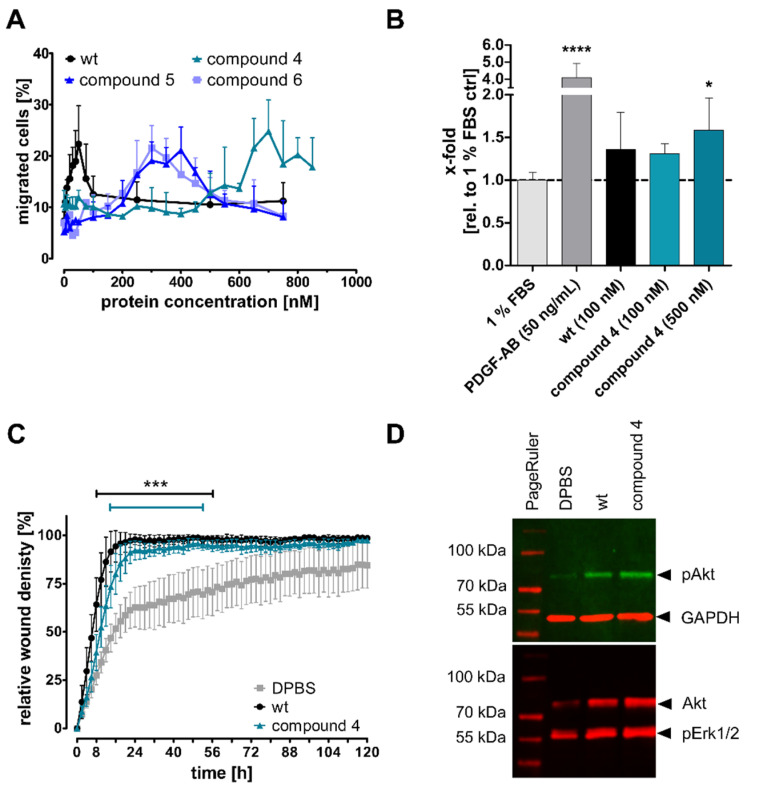
Cellular response of modified CXCL12 variants in solution. (**A**) The migration of Jurkat cells was investigated in a Transwell migration assay to different CXCL12 concentrations. The data are shown as the mean ± SD, *n* ≥ 3. (**B**) Transwell migration assay with primary murine MSCs isolated from C57Bl6/J mice. The cells were incubated in ThinCert™ cell culture insert for 5 h with the respective stimulants. PDGF-AB was used as a positive control, and a medium with 1% FBS was used a as negative control as well as for data normalization. Migrated cells were measured with flow cytometry and total amount of living cells was taken into account; *n* = 2 (three replicates each), mean ± SD. Significance is displayed as **** = *p* ≤ 0.0001; * = *p* ≤ 0.05 compared with 1% FBS. (**C**) Gap closure by keratinocyte migration (HaCaT cells) was made in a scratch assay over five days after pre-treatment with mitomycin C, a potent proliferation inhibitor. The cells were treated with 100 nM CXCL12 wt or compound **4**. The data are presented as mean ± SD, *n* = 3. Significance is displayed as *** = *p* ≤ 0.001 compared with DPBS. (**D**) Representative Western blot analysis of Akt and Erk1/2 migration pathway activation in human fibroblasts. The cells were treated with 100 nM CXCL12 wt or compound **4** for 10 min.

**Figure 3 pharmaceutics-13-01597-f003:**
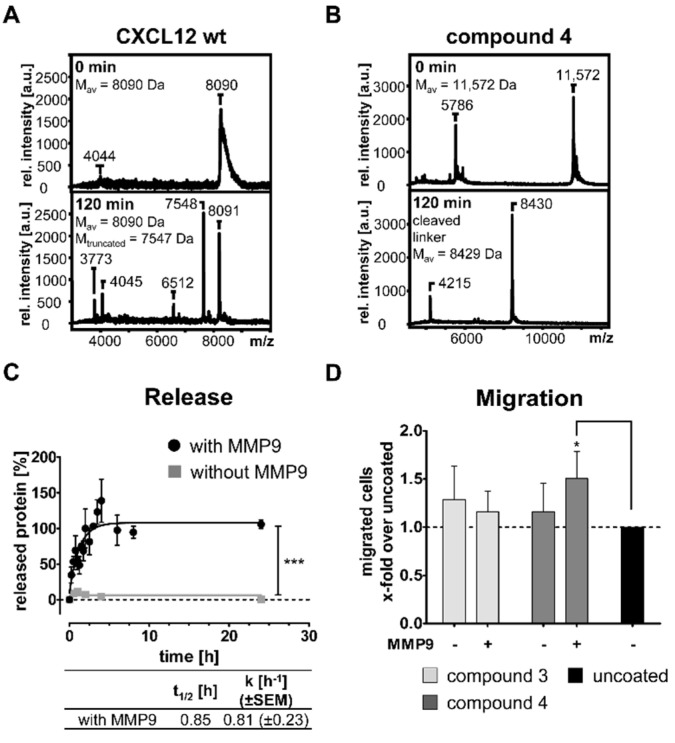
Stabilization and release of modified CXCL12. (**A**) Truncation of native CXCL12 wt by MMP9 and (**B**) for the [S4V]-stabilized compound **4** with MMP9 (0.2 ng/µL), investigated by MALDI-ToF MS analysis. (**C**) Release from coated PCL-*co*-LC scaffolds upon MMP9 incubation of compound **4**. Half-time of release and the rate constant k are given in the corresponding table. Data are shown as mean ± SEM, *n* ≥ 2. Significance is displayed as *** = *p* ≤ 0.001. (**D**) Migration of Jurkat cells in a Transwell migration assay towards gradients of in situ released compound **4** by action of MMP9 from PCL-*co*-LC scaffolds vs. supernatants from non-releasable compound **3**. The data are expressed as mean ± SD, *n* = 3. Significance is displayed as * = *p* ≤ 0.05 compared to uncoated scaffolds.

**Figure 4 pharmaceutics-13-01597-f004:**
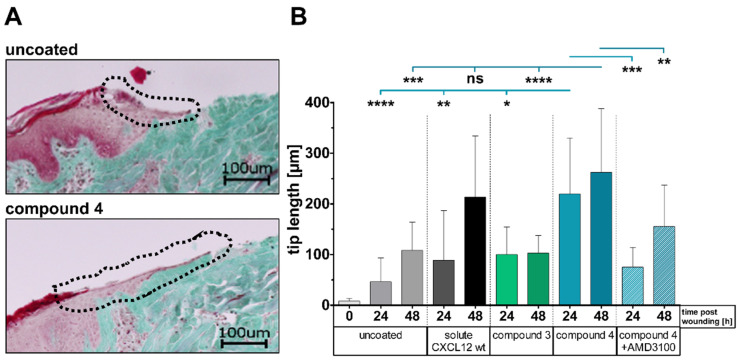
Chemokine release from coated PCL-*co*-LC scaffolds supports epithelial tip formation for wound closure in porcine skin organ culture. Scaffolds were applied for up to 48 h ex vivo on experimental wounds of porcine skin. (**A**) The epithelial tip length was analyzed after Masson–Goldner Trichrome staining of paraffin sections. Representative images of wounds treated with uncoated scaffolds as controls or with scaffolds coated with compound **4**. Scale bar = 100 µm. (**B**) Data on combined tip lengths of both wound sites, shown as mean ± SD, *n* = 3. Significance is displayed as **** = *p* ≤ 0.0001; *** = *p* ≤ 0.001; ** = *p* ≤ 0.01; * = *p* ≤ 0.05; ns = not significant.

**Table 1 pharmaceutics-13-01597-t001:** Overview of generated CXCL12 variants.

A	B	C			D	E		F		
Compound	CXCL12 Variant	Stabilized	Adhesive	Releasable	Monoisotopic Mass [Da]	Elution [%ACN]	Purity [%]	EC_50_ [nM]	pEC_50_ ± SEM	E_max_ [%] ± SEM
wt	CXCL12 wild type	−	−	−	8085.30	37.3	>95	5.4	8.27 ± 0.03	100.0
**1**	[V49A]-CXCL12-SBP(Biotin)	−	+	−	9505.01	37.6	>95	6.2	8.21 ± 0.05	91.5 ± 2.4
**2**	[V49A]-CXCL12-MMPCS-SBP(Biotin)	−	+	+	12,005.05	39.1	>95	126.6	6.90 ± 0.05	93.7 ± 2.8
**3**	[S4V]-[V49A]-CXCL12-SBP	+	+	−	9064.80	37.6	>95	43.3	7.36 ± 0.06	91.1 ± 3.0
**4**	[S4V]-[V49A]-CXCL12-MMPCS-SBP	+	+	+	11,565.29	39.0	>95	54.0	7.27 ± 0.17	84.0 ± 7.5
**5**	[S4V]-[V49A]-CXCL12-GPLS	+	−	−	8427.53	39.0	>95	55.5	7.25 ± 0.07	87.4 ± 2.9
**6**	[S4V]-[V49A]-CXCL12	+	−	−	8073.34	38.3	>95	53.7	7.27 ± 0.06	89.6 ± 2.5
SBP	SBP(Biotin)	−	+	−	1465.75	24.6	>95	nd	nd	0 ± 2.5

Constructs are numbered in column (**A**), and descriptive names are given in column (**B**). Relevant properties of each variant are described in column (**C**), and calculated monoisotopic masses are listed in column (**D**). In column (**E**), the ACN-concentration needed for protein elution is displayed and purity was determined by RP-HPLC on a Phenomenex Aeris Widepore XB-C18 column (200 Å) using linear gradients of 0.1% TFA/H_2_O and 0.08% TFA/ACN. The analytics are included in Appendix A. In column (**F**), the activity of transient transfected COS-7 cells with the CXCR4 and a chimeric Gα_qi_ protein after stimulation with CXCL12 variants for 1 h is presented. IP accumulation was detected by FRET-donor/acceptor-labeled antibodies, and homogenous time resolved fluorescence (HTRF) was measured. The EC_50_, pEC_50_, and E_max_ values are normalized to CXCL12 wt. The data are presented as mean ± SEM, *n* ≥ 3. Visualization is shown in Appendix A. nd = not detectable; wt = wild type; SBP = surface-binding peptide; MMPCS = matrix metalloproteinase cleavage site; ACN = acetonitrile.

## Data Availability

Not applicable.

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
