# Peer review of "Protease-Triggered Release of Stabilized CXCL12 from Coated Scaffolds in an Ex Vivo Wound Model"

_pharmaceutics, 2021, doi:10.3390/pharmaceutics13101597_

Round 1

Reviewer 1 Report

Overview and comments:

The authors present the synthesis of a mussel derived surface-binding peptide coupled with CXCL12 and evaluate its potential as scaffold coating material for skin regeneration. The manuscript presents cellular response of modified CXCL12 variants in solution, evaluation of stabilization & release of modified CXCL12, and evaluation of epithelial tip formation for wound closure in porcine skin organ culture.  

Comments:

It is a well written, and organized manuscript; The experimental design is clear and detailed. It is a preliminary study that will need further experimentation in order to claim the feasibility of the proposed application.

Minor comments:

  1. The authors claim the novel coating synthesized might be a good candidate as scaffold in skin regeneration. Can the authors provide additional data that supports the novel coating’ stability (physical, chemical, and mechanical) when treated under standard clinical protocols of sterilization?

  1. When talking about scaffolding materials and tissue regeneration, pore size (distribution and density) become a critical feature to evaluate. Can the authors data of the scaffold materials with and without coating?

  1. The listed figures below must be replaced by a HD version. When going through the document they seem blurry and distorted.

  1. Figure 1. Conceptional scheme.
  2. Figure 2. Cellular response of modified CXCL12 variants in solution.
  3. Figure 3. Stabilization and release of modified CXCL12.
  4. Figure 4. Chemokine release from coated PCL-co-LC scaffolds supports epithelial tip formation for wound closure in porcine skin organ culture.

Reviewer 2 Report

This manuscript introduces coating the surface of biomaterials with bio-functional proteins to achieve better cell integration. The authors performed appropriately in vitro, ex vivo tests to accomplish better cell integration. The presented data display a successful application of protein-coated biomaterials for skin regeneration. So, it is suitable for publication in the journal "Pharmaceutics" since it has an interesting topic and results. However, it has the following revised parts. They should be checked prior to publication. Followings are recommended for the revision.

Minor revisions:

  1. In all figures, they have to upgrade their resolution.
  2. In figure 4, the number (100 um) upper to the scale bar is not similar in size. Please check it.
  3. In page 5 and section of “Isolation and culture of MSCs”, Please add what kind of plate that you used (e.g. Ultra-low attachment 96-well plate)
  4. In figure 2, A, x-axial, It is recommended to replace the format “c [nM], protein” to “ protein concentration [nM]”

Reviewer 3 Report

The authors describe a very interesting strategy to control the release of chemokines important for the healing of wounds. the paper is well written and the resuslts are convincing and well discussed.

The only suggestion for future studies will be to better define the activity of this scaffold in in vivo models, mainly in diabetic animals. 
